# Oil palm expansion increases the vectorial capacity of dengue vectors in Malaysian Borneo

**Nichar Gregory**[1]*, **Robert M. Ewers**[1], **Arthur Y. C. Chung**[2], **Lauren J. Cator**[1]

**1** Department of Life Sciences, Imperial College London, Silwood Park, Berkshire, United Kingdom, **2** Forest Research Centre, Forestry Department, Sandakan, Sabah, Malaysia

* nicharg@gmail.com

**Data Availability Statement:** All data are available from the Zenodo database (DOI:10.5281/zenodo. 3994260).

## Abstract

Changes in land-use and the associated shifts in environmental conditions can have large effects on the transmission and emergence of mosquito-borne disease. Mosquito-borne disease are particularly sensitive to these changes because mosquito growth, reproduction, survival and susceptibility to infection are all thermally sensitive traits, and land use change dramatically alters local microclimate. Predicting disease transmission under environmental change is increasingly critical for targeting mosquito-borne disease control and for identifying hotspots of disease emergence. Mechanistic models offer a powerful tool for improving these predictions. However, these approaches are limited by the quality and scale of temperature data and the thermal response curves that underlie predictions. Here, we used fine-scale temperature monitoring and a combination of empirical, laboratory and temperature-dependent estimates to estimate the vectorial capacity of *Aedes albopictus* mosquitoes across a tropical forest–oil palm plantation conversion gradient in Malaysian Borneo. We found that fine-scale differences in temperature between logged forest and oil palm plantation sites were not sufficient to produce differences in temperature-dependent demographic trait estimates using published thermal performance curves. However, when measured under field conditions a key parameter, adult abundance, differed significantly between land-use types, resulting in estimates of vectorial capacity that were 1.5 times higher in plantations than in forests. The prediction that oil palm plantations would support mosquito populations with higher vectorial capacity was robust to uncertainties in our adult survival estimates. These results provide a mechanistic basis for understanding the effects of forest conversion to agriculture on mosquito-borne disease risk, and a framework for interpreting emergent relationships between land-use and disease transmission. As the burden of *Ae. albopictus*-vectored diseases, such as dengue virus, increases globally and rising demand for palm oil products drives continued expansion of plantations, these findings have important implications for conservation, land management and public health policy at the global scale.

**Funding:** This study was supported by an Imperial College London Grantham Institute for Climate Change Research Science and Solutions for a Changing Planet doctoral training program (SSCP DTP) studentship awarded to NG, and Sime Darby Foundation funding to the SAFE Project. The funders had no role in study design, data collection and analysis, decision to publish, or preparation of the manuscript.

## Author summary

The large-scale modification of landscapes by humans has contributed to the rise of emerging and re-emerging mosquito-borne diseases. While the association between anthropogenic land-use change and these shifts in disease risk are frequently observed, our understanding of exactly how land-use change mechanistically alters disease risk remains unclear. Changes in local environmental conditions (e.g. temperature) may play an important role, due to the effects on mosquito life-history, but are rarely measured at scales relevant to these small-bodied ectotherms. Here we measure the impact of tropical forest conversion to oil palm plantation on each of the components that determine the potential for mosquitoes to transmit dengue virus (vectorial capacity). Dengue is one of the fastest-growing global infectious diseases, with 100–400 million new infections per year, and whilst *Aedes aegypti* is the dominant vector, the role of *Ae. albopictus* in epidemic outbreaks is of increasing concern. By combining fine-scale temperature data from the field, published temperature responses of *Ae. albopictus* mosquitoes and field data on adult mosquito populations, we show that land-use change from forest to plantation can be expected to increase vectorial capacity by 50%. Our results highlight the need to advance field research into fundamental mosquito ecology, and to more critically evaluate the increased risk of *Aedes*-borne disease in dynamic working landscapes against the benefits of economic development.

## 1. Background

More than 80% of the global population is at risk of being infected with a mosquito-borne disease [1]. Dengue virus, the most prevalent of the mosquito-borne diseases, is estimated to infect 100–400 million people annually [2], and two previously obscure arboviruses, Zika and chikungunya virus, infected over 1 million people between 2016 and 2017 [1]. Climate and environmental change (e.g. land-use) have been implicated as drivers of mosquito-borne disease emergence [3,4] due to the temperature sensitivity of parasite and mosquito traits underlying transmission. However, the mechanisms by which environment influences transmission dynamics remain poorly characterised.

Efforts to identify environmental drivers of disease have historically relied on statistical approaches that link environmental data to disease prevalence, or on occurrence mapping of the mosquito vectors [5,6]. Whilst useful for identifying broad, correlative patterns of disease incidence, these models don't explicitly incorporate biological mechanisms, and are thus limited in their capacity to address non-linear feedback pathways, spatiotemporal heterogeneities and complex transmission dynamics [7]. Mechanistic models that incorporate mosquito and parasite life-history into a mathematical framework for transmission have shown promise. For example, the vectorial capacity model, which incorporates both mosquito and pathogen traits to estimate the total number of potentially infectious bites that would eventually arise from all the mosquitoes biting a human on a single day explicitly [8]. A key limitation of these mechanistic approaches has been a lack of data on the key entomological parameters [9,10]{Mordecai, 2013 #159}.

Mosquitoes are small-bodied ectotherms and their physiology is closely linked to ambient environmental conditions [11]. Temperature affects all components of vectorial capacity: mosquito density, by determining rates of mosquito demographic traits, such as growth, survival and reproduction [12–14], biting rate [15,16], vector competence [17,18], and the pathogen extrinsic incubation period [19,20]. Naturally the vast majority of studies have focused on this

source of variation [21]. Temperature has most recently been incorporated into mechanistic models using thermal performance curves (TPCs) [22–24].

The temperature dependence of mosquito traits is typically assessed under laboratory conditions, where the response of a relevant trait is measured across a range of constant temperatures and yields a curve where performance increases slowly with temperature to a maximum level then rapidly declines [13,25,26]. However, mosquitoes are subject to daily temperature fluctuations in the field, and a growing body of work demonstrates that these variations can produce trait responses that differ to those derived under constant conditions [27,28]

Similarly, whilst increasingly fine-scale environmental data are available (e.g. CHIRTS-daily[29]), the data used to drive transmission models are often coarse in scale, such as average monthly temperatures or broad categories of land-use. These data are unlikely to represent the realised environments of mosquitoes or their pathogens, and the mismatch between the underlying biological processes driving transmission and covariates can have profound effects on predictions for transmission [9,23,30,31].

Fine-scale heterogeneity in landscape structure can significantly increase the disparity between available environmental data and actual environmental conditions [32,33]. For example, the temperature data underpinning many disease transmission models is obtained from the WorldClim2 database, which uses data interpolated from weather stations located in open areas, and has a resolution of approximately 1km. In contrast, vegetation structure mediates microclimates such that temperatures under dense canopy can be ~2–3˚C cooler than in open areas [32,34] and are highly heterogeneous over small spatial scales. These differences are even greater when compared to satellite estimates of local temperatures, with one study finding that within-forest temperatures differ from satellite estimates by 5–10˚C [35]. Similarly, land cover data from satellite-based remote sensing (e.g. Landsat) are typically available at a resolution of 15–60 m, and suffers from high cloud cover in the tropics [36].

Characterising the dynamics of tropical environments is particularly important because they both experience some of the highest rates of anthropogenic land-use change [37] and bear significant and increasing morbidity and mortality due to mosquito-borne diseases [38]. For example, an increase in demand for African oil palm (*Elaeis guineensis*) products has resulted in the dramatic expansion of industrial plantations in the last 20 years [39]. More than 80% of global palm oil is produced in Malaysia and Indonesia [40]. Conversion has been at the expense of either selectively logged or old growth forest [41]. The effects of land-use change on arbovirus transmission in this region remain largely unexplored, despite observations that the abundance of *Aedes albopictus*, an important arbovirus vector, increases following forest conversion to agricultural land [42,43]. This aggressive, day-biting mosquito is native to the forests of Southeast Asia and considered an important vector of the dengue virus and a potential bridge vector for emerging pathogens [44–46]. There has been a dramatic increase in the intensity and magnitude of dengue outbreaks in Malaysia over the last few decades [47,48]. The majority of recent cases were reported from Selangor state, the most populous state in peninsular Malaysia, however, an increasing number of sub-urban and rural cases have been reported [44]. Whilst *Ae. aegypti* has been implicated as the dominant vector in peninsular Malaysia, entomological surveys in dengue outbreak areas in Borneo have found higher abundance of *Ae. albopictus [44,49]*, suggesting that they may play a key role in transmission here. Notably, in 2007 a fifth dengue virus serotype was isolated from an outbreak in the Bornean state of Sarawak. The serotype was identified as a member of a sylvatic lineage of DENV-2, representing the first identification of a sylvatic DENV circulating in Asia since 1975 [50].

We used a mechanistic approach to investigate how tropical forest conversion will affect the vectorial capacity of *Ae. albopictus* mosquitoes. We directly measured *Ae. albopictus* survival, and abundance in forest and oil palm sites. We then measured the effect of thermal

environment on adult gonotrophic cycle length by subjecting females to field thermal conditions in a controlled experiment. These measures were then combined to parameterise a dengue-specific vectorial capacity model to investigate differences in transmission potential between land-use types. We also estimated adult survival and abundance from published TPCs to determine if these predictions differed to our field data. Based on our data, vectorial capacity was estimated to be greater in oil palm plantation than in forest, largely driven by greater mosquito abundance in oil palm. This key difference between land-use types was not predicted by commonly used TPCs. Our results suggest that oil palm conversion may be an important driver of *Aedes*-borne disease emergence.

## 2. Methods

### 2.1 Ethical statement

The use of HLCs was approved by the ethical committee of the Ministry of Health in Malaysia (Approval Number: NMRR-17-3242-39250, Issued: 13 March 2018), and the Imperial College London Research Ethics Committee (ICREC Reference: 17IC3799, Issued: 22/02/17).

### 2.2 Study site

Fieldwork was conducted at sites within the Stability of Altered Forest Ecosystems (SAFE) Project, a large-scale deforestation and forest fragmentation experiment located within lowland dipterocarp forest regions of East Sabah in Malaysian Borneo (4˚33'N, 117˚16'E; [51]. Climate in the region is typically aseasonal [52], with occasional droughts that are often, but not always, associated with the positive phase of ENSO events [53]. The logged forest sites have undergone two rounds of selective logging since 1978, and have a mean aboveground biomass (ABG) of 122.4 t/ha [54]. Logging intensity in this area varies considerably, however most stands are heavily degraded and are characterised by a paucity of commercial timber species, few emergent trees and the dominance of pioneer and invasive vegetation [54]. Oil palm plantations were established in 2006 [51] and are characterised by monocrop stands of closed or nearly-closed canopy oil palm. The plantation sites have considerably lower plant biomass than the forest sites (ABG = 38.1 t/ha;[54]). Mean altitude across the sampling points is 450m (median = 460m, interquartile range = 72m). Further details of the sites are available in the electronic S1 Table).

### 2.3 Fine-scale temperature data

Hourly temperature data were obtained from a separate study conducted during the same period from nine field sites in each of logged forest and oil palm plantations (N = 18; [55]). These data were collected using Data Loggers ERL-USB-2 (LASCAR electronics, Salisbury) secured to small wooden posts at a height of 5 cm, and shaded from rain and direct sunlight with plastic plates suspended approximately 30 cm above each post.

### 2.4 Adult mosquito collection

We directly measured adult mosquito density from four human settlements. Two of these were located within logged forest and two were located within oil palm sites. Host-seeking mosquitoes were collected using the human landing catch (HLC) method, where a collector (NG) sat with their limbs exposed and aspirated mosquitoes as they attempted to feed. Sampling was conducted at each site between February and April in both 2017 and 2018. Collections were carried out for up to three consecutive days, at least twice per site between 10:00 to

12:00. The collector was trained in the HLC method prior to involvement in the project and their health monitored for three weeks following collections.

## 2.5 Estimating mosquito survival

Adult survival was estimated using the proportion of host-seeking females that were found to have laid one or more batches of eggs [56]. Female mosquitoes collected during HLCs were immobilised in a freezer and then had their ovaries removed for observation of the tracheole coils, following Detinova [57]. Nulliparous mosquitoes have tightly coiled ovarioles that become irreversibly distended during the passage of eggs, creating bead-like dilatations. The population of adult females is thus split into two age groups representing young and old individuals. Adult survival ($s$) can then be estimated using the equation:

$$S = M\frac{1}{i_0}$$

where $M$ is the parous rate of the sampled population and $i_0$ is the length of the first gonotrophic cycle [58,59].

To determine gonotrophic cycle length ($i_0$) *Ae. albopictus* larvae were experimentally reared in incubators (Panasonic MLR-350H). Eggs between 2–4 weeks old were hatched in dechlorinated water over a period of 36 hours. First instar larvae were reared at a density of 50 larvae per cup containing ~300mL water, and provisioned with five pellets of fish food (*Tetra Cichlid Colour*) daily. Larvae were exposed to temperature treatments simulating mean hourly diurnal temperatures empirically recorded in either oil palm plantation (mean = 24.8˚C, range = 6˚C) or logged forest (mean = 24.1˚C, range = 4˚C) sampling sites [55] or a control treatment where temperature was held constant (24˚C). Temperature data and rearing protocol are described in further detail in the electronic S1 Table. Upon emergence, male and female adults were housed together to allow for mating, and provisioned with 10% sugar solution *ad libitum*. Females were offered defibrinated horse blood via a Hemotek feeding system three days after emergence. Mosquitoes that did not fully engorge were removed from the fecundity experiments. Mosquitoes that failed to feed were offered a bloodmeal each day until a successful feed was achieved or until four additional days had passed. Blood-fed females were transferred into individual laying tubes (2.9cm x 11.7cm) containing damp filter paper as an egg laying substrate. Papers were observed every day for the presence of eggs, and the length of the pre-bloodmeal period as well as the first gonotrophic cycle were recorded. The first gonotrophic cycle period was used as an estimate of average gonotrophic cycle length when calculating survival [60], as only one bloodmeal was provided.

## 2.6 Estimating mosquito density based on published thermal performance data

We used our fine-scale data to approximate wild mosquito density using a method used in a number of other VBD studies [9,10,22,61]. Here density is estimated using the equation:

$$m = \frac{EFD \cdot_{pEA} .MDR}{\mu^2}$$

where the parameters are the number of eggs laid per female per day (*EFD*), the egg to adult survival probability (*pEA*), the larval development rate (*MDR*), and adult daily rate of mortality ($\mu$) [62]. Trait values were estimated using temperature-dependent trait response curves from Mordecai *et al.* [22], combined with the fine-scale temperature data collected from the field. When necessary, traits were adjusted to the correct metric (e.g. in the original work

characterising thermal performance curves, *EFD* is eggs laid per gonotrophic cycle, *TFD*). Rate summation was then used to incorporate the effects of diurnal temperature fluctuation by estimating trait responses at each hour throughout the day and summing the proportional hourly changes [63,64].

### 2.7 Estimating vectorial capacity

Vectorial capacity was estimated using a dengue specific framework:

$$VC = \frac{m \cdot a^2 \cdot b \cdot c \cdot e^{\frac{-\mu}{REI}}}{\mu}$$  3

where *m* denotes vector to human ratio, *a* is the daily probability of a female mosquito taking a bloodmeal, *b* is the probability of transmission from an infectious mosquito to a susceptible human, *c* is the probability of transmission from an infectious human to susceptible mosquito, *REI* is the rate of extrinsic incubation of the pathogen, and *μ* is the daily rate of adult mortality.

We parameterized this model using a combination of our field-collected data on mosquito life history and abundance (Table 1) and estimates derived from published TPCs. Mosquito density and survival were measured in the field, and female biting rate was taken as the inverse of the gonotrophic cycle length derived from our laboratory experiments [12,22]. Parameters that were not measured in this study (*REI*, *b*, *c*) were estimated using field-collected temperature data and rate summation as described above. We also calculated relative vectorial capacity (*rVC*, the vectorial capacity relative to the vector: human population ratio) for each of the sites [65]. As the mosquito to human ratio (*m*) scales proportionally with vectorial capacity, omitting abundance from the vectorial capacity equation allows us to compare the relative contributions of an individual mosquito inhabiting logged forest and oil palm plantation to transmission.

### 2.8 Sensitivity of vectorial capacity estimates to mortality and biting rate

Vectorial capacity is especially sensitive to variation in adult mosquito survival and biting rate. To explore the impact of uncertainty in survival estimates derived from our field data, we recalculated vectorial capacity for every combination of daily adult mortality rate and of biting rate in the two land-use types at increments of 0.01, while holding all other parameters constant. We used a contour plot to visualize how differences in relative survival affected transmission potential.

**Table 1. Traits and data sources for vectorial capacity parameters [22].**

| Trait | Symbol | Source |
|---|---|---|
| Vector density | *m* | Measured directly in field collections |
| Daily probability of a female mosquito taking a bloodmeal | *a* | Calculated from laboratory experiments |
| Probability of transmission from an infectious mosquito to a susceptible human | *b* | Mordecai *et al.* (2017) [22] temperature-dependent estimates combined with field-collected temperature data |
| Probability of transmission from an infectious human to a susceptible mosquito | *c* | Mordecai *et al.* (2017) [22] temperature-dependent estimates combined with field-collected temperature data |
| Extrinsic incubation rate of the pathogen | *REI* | Mordecai *et al.* (2017) [22] temperature-dependent estimates combined with field-collected temperature data |
| Daily rate of adult mortality | *μ* | Estimated from field parity assessments and gonotrophic cycle length determined in laboratory experiments |

### 2.9 Statistical analysis

All analyses were carried out in R Version 3.5 [66]. A generalised linear negative binomial model (link = log; R package MASS; [67], with land-use type set as a predictor was used to compare the number of host-seeking mosquitoes collected in HLCs between oil palm plantations and logged forest sites. Sampling site and year were initially nested as random effects, but neither was significant in the model so were excluded from the final model. To calculate survival, parous rates were compared between land-use type and sampling years using a generalised linear model (family = Gamma, link = "log"). Laboratory data on gonotrophic cycle length were compared using the same model structure with the length of the first gonotrophic cycle in days set as the response variable and experimental treatment (forest or plantation microclimatic conditions) set as the predictor. Adult survival was then calculated from mean parous rate and gonotrophic cycle using Eq 1.

Mean trait estimates for each land-use type were then used to calculate vectorial capacity at each location for which there was temperature data (N = 18). If the point estimates for each trait did not differ significantly between land-use types, we took the conservative approach of using the average trait value across both land-use types. The *b*, *c*, and *REI* values estimated from Mordecai *et al.* [22] were compared between land-use types using Mann-Whitney U tests. The effect of land-use on vectorial capacity estimates was then explored using linear models, with land-use and sampling year as predictors. Relative vectorial capacity estimates were compared using a Mann-Whitney U test.

## 3. Results

### 3.1 Human landing catches

A total of 276 *Ae. albopictus* mosquitoes were collected from 46 HLC surveys in 2017 and 22 surveys in 2018. Male mosquitoes typically arrived first, and made up 19–58% of the collections at each site (Table 2). However, this is likely an underestimate of the number of males swarming as only those that landed on the collectors were counted. The number of female mosquitoes collected per day ranged from 1–6 in logged forest and 1–15 in plantations, and were not significantly different between sampling years (GLM, RR = 0.01, df = 64, 95% CI: -0.29,0.31, p = 0.9). Overall, human landing catches conducted in plantations yielded an average 1.5 times more mosquitoes per session than those at logged forest sites (GLM, RR = 0.42, df = 64, 95% CI: 1.17, 2.0, p = 0.002). The mean number of mosquitoes collected per session was 3.59 ± 0.27 (SE) in logged forest, and 5.48 ± 0.65 (SE) in plantation sites. The proportion of parous mosquitoes was not significantly different between land-use types (GLM, $F_{1,6}$ = 0.2,

**Table 2. Human landing catch data.** Sampling sites are listed along with the number of sampling days per site (n). For the proportion of parous females, *n* indicates the total number of females for which parity status could be determined.

| Year | Land-use type | Sampling site (n) | Total number of females/males | Mean number of females per day (95% CI) | Proportion parous (n) |
|------|---------------|-------------------|-------------------------------|------------------------------------------|------------------------|
| 2017 | Oil palm | OPK1 (11) | 76/22 | 6.91 (4.1, 9.7) | 0.90 (44) |
|      |          | OPSB (8) | 32/25 | 4.57 (2.5, 5.5) | 0.75 (18) |
|      | Logged forest | SAFE (18) | 58/14 | 3.22 (2.1, 7.0) | 0.80 (41) |
|      |          | SWML (5) | 28/13 | 4.00 (2.7, 5.3) | 0.82 (11) |
| 2018 | Oil palm | OPK1 (6) | 28/38 | 4.67 (2.3, 7.0) | 0.70 (20) |
|      |          | OPSB (5) | 23/9 | 5.75 (0.03, 12) | 0.72 (18) |
|      | Logged forest | SAFE (6) | 38/7 | 4.75 (3.5, 7.0) | 0.76 (37) |
|      |          | SWML (5) | 9/10 | 2.25 (0.7,3.8) | 0.89 (9) |
|      | **TOTAL** | **64** | **292/138** | -- | **(198)** |

p = 0.69) or sampling years (GLM, $F_{1,5}$ = 0.98, p = 0.37). Parity rates were highly variable within sites, with both the highest value in 2017, and the lowest value in 2018, observed at the same site (0.90 and 0.70, respectively).

### 3.2 Adult survival and temperature-dependent transmission

Gonotrophic cycle length did not differ between logged forest and oil palm plantation treatments (Mann-Whitney U, Z = -0.99, n = 83, *p* = 0.33), averaging 8.24 ± 3.2 (±SD) days. As neither gonotrophic cycle length nor parity rate differed significantly between land-use types, the median value across all sites was used to estimate adult survival (S) in both land-use types. Daily rate of adult mortality ($\mu$) was then taken as 1 –S, and estimated to be 0.03 ± 0.006 (SE) for all sites, corresponding to an estimated adult lifespan of 33.3 ± 7 (SE) days. Biting rate was taken as the inverse of the median gonotrophic cycle length, and set as 0.12 for all sites.

Temperature data indicated that although mean daily temperatures did not differ significantly between land-use types during the sampling period, daily temperature fluctuations were greater in oil palm plantations than in logged forest [55]. Despite warmer temperatures in plantation sites, none of the parameters estimated using the temperature-dependent responses from Mordecai *et al.* [22] differed significantly between the land-use types for either sampling year (Wilcox, Z = 0.43, *p* = 0.663; Fig 1). The probability of dengue transmission from an infectious mosquito to a susceptible human (*b*) was estimated to be 0.501 ± 0.01 in logged forest and 0.503 ± 0.012 (mean ± SE) in oil palm. Human to mosquito transmission probability (*c*) was estimated to be 0.762 in both logged forest and oil palm, with marginally greater variation in the latter (logged forest SE = 0.005, oil palm SE = 0.008). The rate of extrinsic incubation (*REI*) was also the same in logged forest and oil palm, and estimated to be 0.155 ± 0.01.

Vectorial capacity was found to be 1.5 times greater in oil palm plantations (1.51 ± 0.03, mean ± SE) than in logged forest (0.99 ± 0.01, mean ± SE; F(1,4) = 61, $\beta$ = 0.50, *p* < 0.001) and did not differ significantly between sampling years ($\beta$ = -0.005, *p* = 0.93). The higher vectorial capacity in oil palm plantations was driven by greater mosquito density (*m*) at plantation sites, as *rVC* estimates did not differ significantly between land-use types (Wilcox, W = 6.5, Z = 0.06, *p* = 0.77).

### 3.3 Effects of adult mortality and biting rate on vectorial capacity

We found that holding all other parameters constant, if adult mortality was equal in logged forest and oil palm plantations, vectorial capacity would always be higher in plantations (Fig 2A), with the difference increasing exponentially with decreasing mortality rate. Given field estimates of adult mortality in oil palm plantations ($\mu$ = 0.03), vectorial capacity in logged forest could only exceed that in plantations if mortality in forests was lower than 0.02. This would translate to an adult lifespan of 50 days, compared to 33 days in oil palm. We note that changing mortality may also be expected to impact mosquito density, however as we derived *VC* from direct field observations of density, we did not include this feedback in our analysis. Calculating *rVC* with equal mortality between the sites demonstrates that *rVC* will not differ significantly between the land-use types within the range of most mosquito lifespans (µ < 0.007, or a lifespan < 140 days; Wilcox, *W* = 42180, Z = -21.2, *p* = 0.72). Similarly, we found that given equal biting rates in forest and plantation, vectorial capacity will always be greater in the latter (Fig 2B). Given our biting rate estimate of 0.125, or once every 8 days, vectorial capacity in forest will only exceed that in oil palm at 0.23, or a frequency of every four days. When calculated from thermal performance curves, biting rates were estimated to increase by 50% compared to estimates derived from laboratory data on gonotrophic cycle length, and did not differ significantly between land-use types (Wilcox, Z = -1.55, p = 0.13). Holding all other

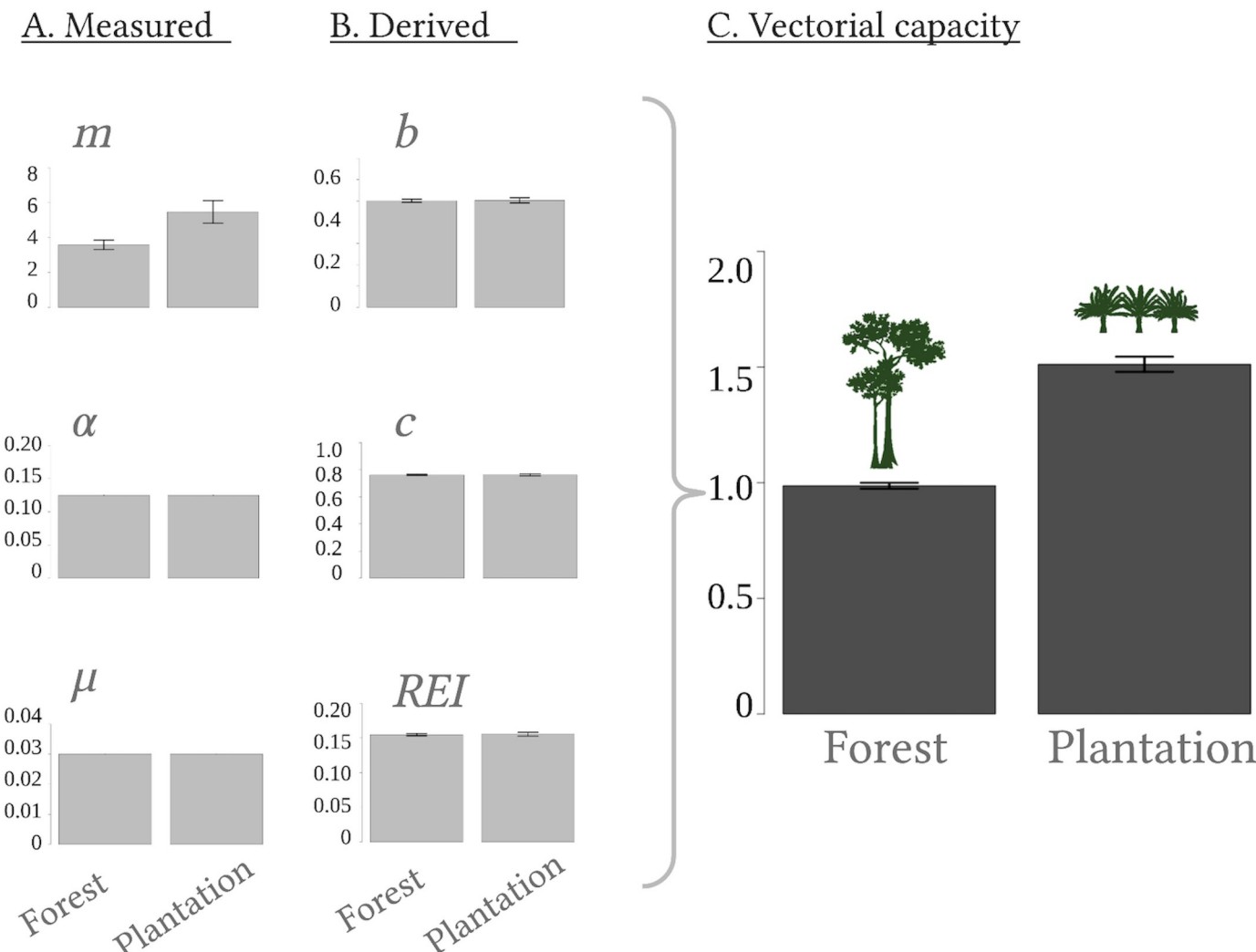

**Fig 1. Effects of land-use on vectorial capacity.** Estimates derived from temperature-dependent trait responses and data measured in the field are shown for each parameter underlying vectorial capacity (*VC*) for 2017. These are vector to human ratio (*m*), daily probability of a female mosquito taking a bloodmeal (*a*), the daily rate of adult mortality (*μ*), the probability of transmission from an infectious mosquito to a susceptible human (*b*), the probability of transmission from an infectious human to susceptible mosquito (*c*), and the rate of extrinsic incubation of the pathogen (*REI*). Created with BioRender.com.

parameters constant, we found that this increase in biting rate resulted in a four-fold increase in vectorial capacity for both logged forest and oil palm plantation sites.

## 4. Discussion

The advantages of using mechanistic models for understanding the impacts of environmental change on mosquito-borne disease dynamics have been demonstrated in a number of studies [9,10,61]. A key limitation to the development of these frameworks has been a lack of field data for model parameterization. By integrating fine-scale field data and temperature-dependent estimates from the literature into a mathematical framework for transmission potential, our study furthers progress in this area in three key ways.

First, in contrast with studies that use coarse, aggregated temperature data or that simulate diurnal temperature to investigate the impacts of environmental change on mosquito traits, we measured fine-scale temperature at field sites where mosquito data were collected. Second,

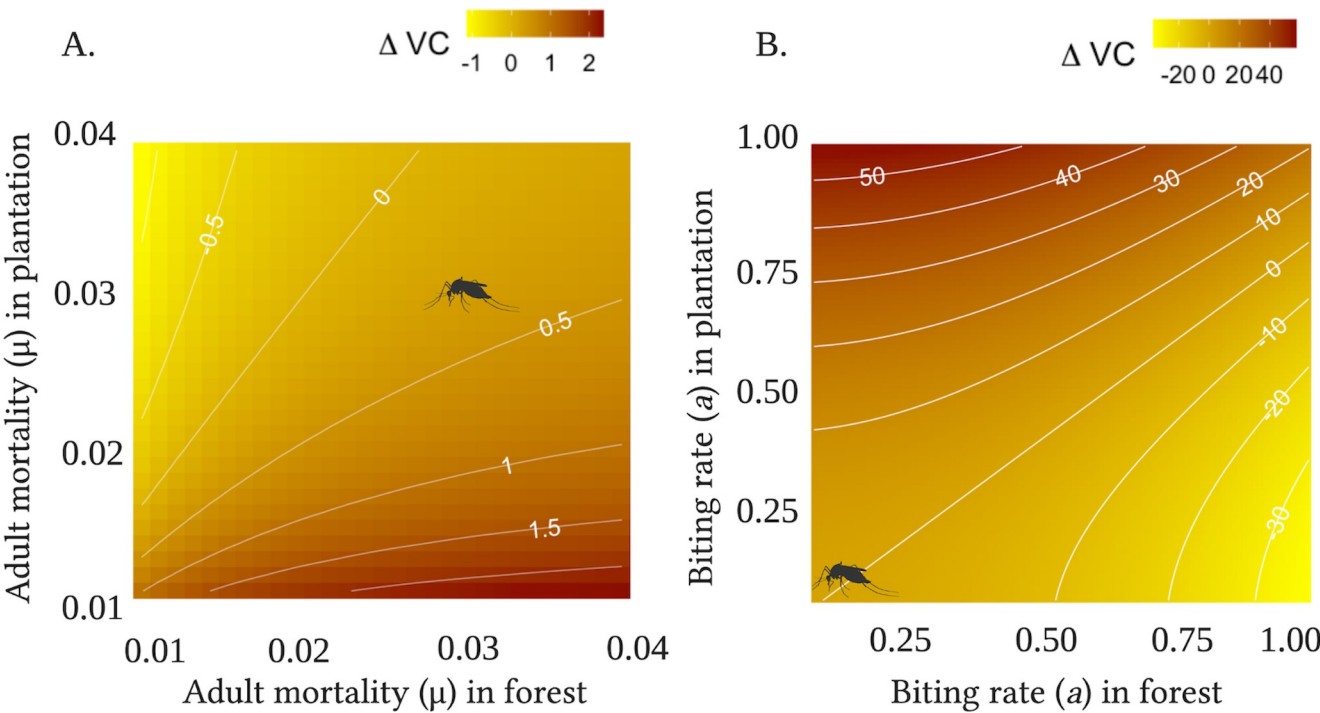

**Fig 2. Effects of adult mortality (A) and biting rate (B) on vectorial capacity.** Contour plot of differences in vectorial capacity estimates between oil palm plantations and logged forest, with positive values of ΔVC denoting higher vectorial capacity in oil palm plantation and negative values denoting higher vectorial capacity in logged forest. As you move along the x-axis, the respective traits increase in logged forest, and the same applies for oil palm plantation along the y-axis. The mosquito icon indicates the value of the trait observed from the empirical data. Created with Biorender.com.

we account for non-linear thermal trait responses and incorporate diurnal temperature fluctuations in trait estimates. Finally, we parameterize our vectorial capacity model using field data on two crucial and challenging to measure parameters: adult abundance and survival.

We estimated the vectorial capacity of *Ae. albopictus* mosquitoes in oil palm plantations to be 1.5 times that of those in logged forests; a result driven by a greater mosquito abundance in plantations. Although we acknowledge small sample size, the observation that *Ae. albopictus* abundance increases in disturbed habitats is supported by previous empirical work conducted in the region [42,43,68], and reflects the well-established ecological plasticity of the species [45]. This increased abundance was not estimated from TPCs, as diurnal temperatures did not differ sufficiently between habitat types to produce divergent responses in the underlying demographic traits. However, our temperature data contrast with a number of other studies, which have found mean temperatures in plantations to be up to 4°C warmer than in logged forests [34,69]. Microclimates vary significantly through the forest to plantation conversion timeline, and as plantations mature, leaf litter and canopy cover increases, buffering microclimates [69]. Our observations that mean temperatures did not differ significantly between sites is thus likely due to selection of older plantations and significantly degraded forest, and it's possible that using temperature data from a wider range of forest and plantations sites to drive our model may have produced different estimates for abundance and vectorial capacity.

An additional consideration is that, although the broad temperatures sampled in this study represent the temperatures experienced by mosquitoes more accurately than those derived from weather station and remotely sensed data, in reality an even wider range of microclimates are available at fine spatial scales [70]. Behavioural avoidance of unsuitable temperatures by mosquitoes may further mediate the relationship between mosquito traits and ambient

temperature. Although typically constrained to small bodies of water, mosquito larvae may move up and down between warmer and cooler layers of water, and adults may simply move away from extreme temperatures [71]. Experimental studies that characterize fine-scale behavioural thermoregulation of mosquitoes remain limited, however a study of *Anopheles gambiae* found that, given access to range of temperatures, almost half of the mosquitoes consistently selected resting sites within 24–27˚C, suggesting some capacity for behavioural thermoregulation [72]. Further experimental work would contribute significantly to our understanding of their realized thermal niche. In the meantime, assessing the effects of temperature on mosquito traits may thus benefit from a temperature envelope approach, whereby the distribution of temperatures are used to evaluate trait responses rather than a single measure per unit time [33].

The characterisation of *Ae. albopictus* thermal responses has advanced mechanistic efforts to understand temperature-pathogen transmission relationships, however an important caveat to consider is that these relationships are produced under constant laboratory conditions, using a variety of laboratory-adapted mosquito strains [22]. Key features of thermal responses curves are often adapted to local climatic conditions [73], and curves aggregated across a range of different mosquito strains, as well as number of laboratory generations, may obscure fine-scale variation in trait responses. In one study, *Ae. albopictus* adults collected across an urbanization gradient in Malaysia exhibited considerable variation in survival, with urban strains surviving approximately one week longer than rural strains, and two weeks longer than laboratory strains under standardised conditions [74]. In a previous study conducted at these same field sites, we found that differences in daily temperature fluctuations resulted in faster larval development rates in oil palm plantations [55], an effect not predicted by published TPCs. Accurately characterising thermal response curves is particularly important given their asymmetric nature. A temperature higher than the thermal optimum depresses fitness more than a temperature displaced an equivalent amount below the thermal optimum [75], meaning that even small shifts in the shape and distribution of key features (e.g. thermal optimum) can dramatically alter how temperature fluctuations are predicted to impact trait performance. This explains the similarity of our temperature-dependent trait estimates, as the daily temperature fluctuations at our sites are small relative to those defining the operational range of *Ae. albopictus*. For example, the thermal optimum for biting rate is approximately 33˚C [22], which is warmer than the maximum temperatures in both oil palm and logged forest sites. Below the thermal optimum, temperature-dependent trait performance approximates a linear response, thus, small differences in daily temperature fluctuation are not sufficient to produce significant differences in estimated trait performance.

By focusing solely on temperature-dependent responses, our vectorial capacity estimates ignore some potentially important sources of socio-ecological variation between land-use types. For example, as forest conversion to oil palm plantation reduces invertebrate diversity and abundance [76,77], predation and competition may also be reduced. Additionally, the homogenisation of vegetation associated with monoculture may reduce breeding habitats by removing trees, but may also increase the availability of artificial breeding sites due to increases in human activity. The plantation and forest sites sampled in this study were all characterised by labour-intensive economies, but the level of investment in housing infrastructure and public services was considerably higher in the former. Variation in water storage, waste management and vector control would all have impacted mosquito abundance, but characterising these factors were beyond the scope of this study. Additionally, we did not collect data on human density, however it was likely higher in plantations than in logged forest and may contribute to the higher mosquito density there.

Although we can only speculate as to the relative impact of these factors, our empirical data suggest that at least one key trait, survival, does not differ significantly between land-use types.

In our modest sample of host-seeking female mosquitoes, we did not observe differences in parity rates among habitat types. Overall parity rates were high across the sites, which we cautiously interpret as indicative of older host-seeking mosquito populations. However, when we used gonotrophic cycle length to calculate lifespan, we estimated that mosquitoes only live for approximately 33 days. We believe this may be an overestimate for two reasons: 1) although our data did not indicate an effect of temperature fluctuations on gonotrophic cycle length, a number of laboratory and field studies have found this trait to be temperature sensitive [12,16,27], and 2) we were only able to measure the first gonotrophic cycle length, which can be twice as long as subsequent cycles [78]. Shorter gonotrophic cycles would affect vectorial capacity estimates by decreasing calculated survival and increasing biting rate. For example, halving the gonotrophic cycle would result in a 95% decrease in vectorial capacity due to effects on the probability of surviving the extrinsic incubation period. However, in this scenario biting rate would increase by 50%. Together these changes in survival and biting rate estimates would reduce vectorial capacity by ~52%. Though ubiquitous, the use of gonotrophic cycle length to infer biting rates introduces an additional source of uncertainty, as aedine mosquitoes are frequently observed taking multiple bloodmeals within a cycle [79–81]. As biting rate is raised to the power of 2 in the vectorial capacity equation, small changes in this parameter can result in relatively large changes in transmission intensity. Sensitivity analyses suggest that our results are robust to uncertainties in survival and biting rate estimates, however further sampling is needed to understand how these traits vary over space and time.

Despite the dominant role of oil palm plantation expansion on deforestation in Malaysia's Borneo states, few studies have explicitly investigated the impacts on mosquito vectors. To our knowledge, this study represents the first attempt to mechanistically characterise the effects of tropical forest conversion to agriculture on the vectorial capacity of *Ae. albopictus*. Our finding that vectorial capacity increases in plantations highlights the need to more critically evaluate the increased risk of *Aedes*-borne disease against the benefits of economic development. Borneo is a region that continues to experience rapid urbanisation, with many urban centers surrounded by forest, or strongly connected to forest via logging and palm oil transport routes. The region is endemic for all four dengue serotypes, and has experienced an increasing number of dengue cases in recent years. In 2018, a total of 3,423 dengue cases were reported from Sabah, a substantial increase from the 2,560 cases in the year previous [44], and Chikungunya has also recently re-emerged in the region [82]. *Ae. albopictus* is suspected to be the dominant vector of both arboviruses in the region [44]. Increasing urbanisation in Borneo may present an additional risk to human health, in that human exposure to circulating sylvatic arboviruses likely increases at the forest interface. The evolutionary histories of dengue and Chikungunya viruses represent a cautionary tale, highlighting the potentially devastating consequences of disease spillover from sylvatic to human cycles [83]. Given their propensity for zoophily, their competence for a number of arboviruses, and their capacity to thrive in both converted and forested areas, *Ae. albopictus* is an obvious candidate for facilitating contacts with enzootic cycles and increasing the risk of spillover events. In Borneo, this has already occurred with anopheline vectors and the simian malaria, *Plasmodium knowlesi*, for which disease risk is closely linked to proximity to forest [84,85], and to a lesser extent, for some sylvatic dengue strains [86]. The effects of land-use change on vector-borne disease risk will be an emergent property of interactions between host and vector ecology, behaviour, physiology and immunity and their idiosyncratic responses to change.

Current understanding of mosquito biology stems largely from studies evaluating specific control interventions, which has left significant knowledge gaps in fundamental mosquito ecology [87]. Mechanistic models, which will play a crucial role in predicting disease transmission dynamics under changing land-use and climate depend heavily on the accuracy of the underlying parameters. As such, there is a need to invest in studies that characterise mosquito

ecology, as well as evaluate how they vary in space and time [7]. This is particularly important for many *Aedes*-borne pathogens, for which disease control is currently entirely dependent on management of complex and dynamic vector populations.

## Supporting information

**S1 Methods. Human landing catch site selection.**
(DOCX)

**S2 Methods. Laboratory rearing protocol.**
(DOCX)

**S1 Table. Temperature schedules of incubators used for larval rearing.** Temperatures were set to change every hour. Minimum ($^*$) and maximum ($_\Delta$) temperatures for each treatment are indicated once in each column.
(DOCX)

## Acknowledgments

Permission to conduct research was provided by the Sabah Biodiversity Council, Yayasan Sabah, Maliau Basin Management Committee (Licence Ref. No.: JKM/MBS.1000-2/2 JLD.4 (174)), and the Southeast Asia Rainforest Research Partnership (SEARRP). Logistical support was provided by the SAFE project coordinator, Ryan Gray, and the fieldwork support by the SAFE project field staff.

## Author Contributions

**Conceptualization:** Nichar Gregory, Robert M. Ewers, Lauren J. Cator.

**Data curation:** Nichar Gregory.

**Formal analysis:** Nichar Gregory.

**Methodology:** Nichar Gregory, Robert M. Ewers, Lauren J. Cator.

**Writing – original draft:** Nichar Gregory, Lauren J. Cator.

**Writing – review & editing:** Robert M. Ewers, Arthur Y. C. Chung.

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
