## [Decision Letter · Decision Letter 0]

1 Jul 2021

Dear Dr Gregory,

Thank you very much for submitting your manuscript "Oil palm expansion increases the vectorial capacity of dengue vectors in Malaysian Borneo" for consideration at PLOS Neglected Tropical Diseases. As with all papers reviewed by the journal, your manuscript was reviewed by members of the editorial board and by several independent reviewers. The reviewers appreciated the attention to an important topic. Based on the reviews, we are likely to accept this manuscript for publication, providing that you modify the manuscript according to the review recommendations. 

Sincerely,

Emily Gurley

Deputy Editor

Reviewer's Responses to Questions

**Key Review Criteria Required for Acceptance?**

**Methods**

-Are the objectives of the study clearly articulated with a clear testable hypothesis stated?

-Is the study design appropriate to address the stated objectives?

-Is the population clearly described and appropriate for the hypothesis being tested?

-Is the sample size sufficient to ensure adequate power to address the hypothesis being tested?

-Were correct statistical analysis used to support conclusions?

-Are there concerns about ethical or regulatory requirements being met?

Reviewer #1: Yes

Reviewer #2: See attached marked-up PDF and narrative.

**Results**

-Does the analysis presented match the analysis plan?

-Are the results clearly and completely presented?

-Are the figures (Tables, Images) of sufficient quality for clarity?

Reviewer #1: Yes

Reviewer #2: See attached marked-up PDF and narrative.

I would like to see a greater, more direct statement that the effect is due to density.

**Conclusions**

-Are the conclusions supported by the data presented?

-Are the limitations of analysis clearly described?

-Do the authors discuss how these data can be helpful to advance our understanding of the topic under study?

-Is public health relevance addressed?

Reviewer #1: Yes

Reviewer #2: Yes.

**Editorial and Data Presentation Modifications?**

Reviewer #1: n/a

Reviewer #2: See attached marked-up PDF and narrative.

**Summary and General Comments**

Reviewer #1: This is an outstanding and important paper that combines field data, laboratory experiments, and mathematical models to mechanistically dissect the impact of palm oil plantations, relative to logged forests, on vectorial capacity of Aedes albopictus for arboviruses like dengue. I have relatively few minor comments on this overall excellent paper.

One important innovation of this study is that the authors estimated mortality rates and mosquito abundance directly in the field. These have previously been major unknowns because they were derived from laboratory estimates, and how well they applied to the field was uncertain.

The equation for m (eq. 2) uses the parameter EFD = eggs per female per day. However, in the original work characterizing the thermal performance curves (Mordecai et al. 2017), the trait estimated was TFD = eggs laid per female per gonotrophic cycle, so the substitution EFD = TFD*a (where a is 1/oviposition cycle length; see supplementary Table A) was made to calculate m for Aedes albopictus. Please note this, and correct the m calculation if needed.

Regarding the calculation of the mortality rate, mu, note that in the previous work that derived the thermal performance curves (Mordecai et al. 2017), mu was defined as a mortality rate (units of 1/day) rather than a probability, as stated in line 305. Based on the calculation of mu as 1/lifespan, I believe that the calculation was correct, just the description of the parameter as a probability is not. I didn’t understand how the calculation of 33.3 +/- 57 days was estimated, since if the mortality rate is 0.03 +/- 0.006, that gives the range in mortality rates of 0.036 (lifespan = 27.8 days) to 0.024 (lifespan = 41.7 days).

In line 197 it says that temperature variation was higher in logged forest than in oil palm plantations, but in lines 316-317 it says the opposite. Which is correct? Even though temperature variation turned out to not play an important role in the mosquito traits, it is still important to clarify this point.

Minor comments:

The Abstract and the Author Summary only mention Aedes albopictus and vectorial capacity, but not which diseases are actually at risk of being transmitted in this system. Especially if this study is motivated by an observation of increased disease incidence in people that contact plantations, it is worth mentioning what disease(s) are important here.

Line 29: “predications” should be “predictions”

Line 48: typo in “anthropogenic”

There are several locations throughout the manuscript where the citations aren’t in the PLOS NTD numbered format.

Lines 141 and 495: “Aedes” should be italicized

Signed, 

Erin Mordecai

Reviewer #2: See attached marked-up PDF and narrative.

PLOS authors have the option to publish the peer review history of their article (what does this mean?). If published, this will include your full peer review and any attached files.

Reviewer #1: Yes: Erin Mordecai

Reviewer #2: No

Figure Files:

Data Requirements:

Reproducibility:

References

---

## [Decision Letter · Decision Letter 1]

2 Feb 2022

Dear Dr Gregory,

We are pleased to inform you that your manuscript 'Oil palm expansion increases the vectorial capacity of dengue vectors in Malaysian Borneo' has been provisionally accepted for publication in PLOS Neglected Tropical Diseases.

Before your manuscript can be formally accepted you will need to complete some formatting changes, which you will receive in a follow up email. A member of our team will be in touch with a set of requests. In addition, one reviewer has suggested some very minor revisions which are detailed at the end of this email.

Best regards,

Emily Gurley

Deputy Editor

Reviewer's Responses to Questions

**Key Review Criteria Required for Acceptance?**

**Methods**

-Are the objectives of the study clearly articulated with a clear testable hypothesis stated?

-Is the study design appropriate to address the stated objectives?

-Is the population clearly described and appropriate for the hypothesis being tested?

-Is the sample size sufficient to ensure adequate power to address the hypothesis being tested?

-Were correct statistical analysis used to support conclusions?

-Are there concerns about ethical or regulatory requirements being met?

Reviewer #1: Methods meet the standard for acceptance

**Results**

-Does the analysis presented match the analysis plan?

-Are the results clearly and completely presented?

-Are the figures (Tables, Images) of sufficient quality for clarity?

Reviewer #1: Results meet the standard for acceptance

**Conclusions**

-Are the conclusions supported by the data presented?

-Are the limitations of analysis clearly described?

-Do the authors discuss how these data can be helpful to advance our understanding of the topic under study?

-Is public health relevance addressed?

Reviewer #1: Conclusions meet the standards for acceptance

**Editorial and Data Presentation Modifications?**

Reviewer #1: (No Response)

**Summary and General Comments**

Reviewer #1: This is an excellent paper, and the revisions have improved it and resolved my questions about the methods. Just a few minor comments.

Line 23: should be “Mosquito-borne diseases”

Line 54: missing word, such as “the transition from”

Equation 3: there should not be a negative sign in the denominator

Line 239: extra comma before the period

Line 254: daily probability of survival should be daily adult mortality rate

Line 469: should be “sensitivity analyses suggest”

PLOS authors have the option to publish the peer review history of their article (what does this mean?). If published, this will include your full peer review and any attached files.

Reviewer #1: **Yes: **Erin Mordecai

---

## [Editor Report · Acceptance letter]

8 Mar 2022

Dear Dr Gregory,

We are delighted to inform you that your manuscript, "Oil palm expansion increases the vectorial capacity of dengue vectors in Malaysian Borneo," has been formally accepted for publication in PLOS Neglected Tropical Diseases.

Best regards,

Shaden Kamhawi

co-Editor-in-Chief

Paul Brindley

co-Editor-in-Chief
